# Fast Exploration with Simplified Models and Approximately Optimistic Planning in Model-Based Reinforcement Learning

## Abstract

Humans learn to play video games significantly faster than the state-of-the-art reinforcement learning (RL) algorithms. People seem to build simple models that are easy to learn to support planning and strategic exploration. Inspired by this, we investigate two issues in leveraging model-based RL for sample efficiency. First we investigate how to perform strategic exploration when exact planning is not feasible and empirically show that optimistic Monte Carlo Tree Search outperforms posterior sampling methods. Second we show how to learn simple deterministic models to support fast learning using object representation. We illustrate the benefit of these ideas by introducing a novel algorithm, Strategic Object Oriented Reinforcement Learning (SOORL), that outperforms state-of-the-art algorithms in the game of *Pitfall!* in less than 50 episodes.

## 1 Introduction

The coupling of deep neural networks and reinforcement learning has led to exciting advances, enabling reinforcement learning agents that can reach human-level performance in many Atari2600 games (Mnih et al., 2015; 2016; Silver et al., 2017; Hessel et al., 2017). However, such agents typically require hundreds of millions of time steps to learn to play well. This is in sharp contrast to people, who typically learn to play Atari games within a few episodes (Lake et al., 2017). Prior work on human learning for Atari suggests that people may be systematically building models of the reward and dynamics in the domain and using those to plan efficiently (Tsividis et al., 2017; Dubey et al., 2018).

Given that human learners seemingly perform model-based RL very quickly, we are motivated to consider alternative approaches to current model-based RL, which often involves building complex predictive deep neural networks from scratch. Deep neural networks can require a large amount of data to accurately train, and performing perfect planning with those models is computationally expensive. Indeed, while there is a considerable amount of theoretical work done on tabular model-based reinforcement learning that suggests model-based approaches can be provably sample efficient (Dann et al., 2017; Brafman & Tennenholtz, 2002; Strehl & Littman, 2008), there is much less success in using model-based reinforcement learning in extremely large environments, in part due to the challenges of learning simple accurate models in these domains.

Instead, in this paper, we investigate two issues in leveraging model-based RL to speed learning in large domains. First we explore how to perform approximate planning using models during model-based RL to support exploration, assuming models are given. Models can facilitate deep exploration, since one key benefit of using models is that it is often easier to quantify uncertainty in those models, and perform planning in a way to guide the agent towards exploring and reducing that uncertainty. Recent work has suggested the benefit of Thompson Sampling methods over optimism methods for reinforcement learning (Osband & Van Roy, 2016), and, indeed, empirically Thompson Sampling methods have done very well in contextual bandits and small state space MDPs where it is possible to plan exactly. However, to our knowledge, there has not been an investigation of how these popular approaches scale to larger domains where it is computationally prohibitive to perform exact planning. We introduce optimistic MCTS, a variant of the popular planning technique Monte

Carlo Tree Search, and find that optimism here outperforms other approaches in several simulation experiments when planning can only be done approximately.

Second, while our first investigation explores how to plan to encourage exploration given a model, we further investigate how to choose the model class to support computational tractability and learning efficiency. Here we propose to learn object-oriented deterministic models of the domain. People may leverage and test models of object interactions during video game learning (Tsividis et al., 2017; Dubey et al., 2018), and object-oriented learning has the nice benefit that data from all similar objects can be pooled when building a model. Prior work has shown that object-oriented model-based reinforcement learning can yield provably efficient learning and scale to larger domains (Diuk et al., 2008). Deterministic models offer an additional benefit over object-oriented models–they require less data to train (since one does not have to model a large stochastic distribution of outcomes) and they have additional benefits for planning, reducing the branching factor of next states to 1. Indeed past work (Diuk et al., 2008) also proposed learning deterministic object-oriented models. In contrast, we investigate how to learn the transition model and temporal abstraction to make the world appear deterministic. In other words, we assume a candidate set of possible transition model classes and temporal abstractions, and perform model selection to select a level of temporal abstraction and model that deterministically predicts the outcome of an action.

We illustrate the potential benefit of these ideas by introducing an object oriented algorithm that uses prior knowledge of model classes and object representation, and show that our algorithm can learn to achieve positive reward in the notoriously difficult Atari game *Pitfall!* within 50 episodes. Almost no RL methods have achieved positive reward on *Pitfall!* without human demonstrations, and even with demonstrations, such approaches often take hundreds of millions of frames to learn (Aytar et al., 2018; Hester et al., 2017). In contrast to demonstrations, we assume two forms of prior knowledge–a predefined object representation and a class of potential model features. Computer vision is rapidly advancing to the state that soon we will be able to easily extract objects from even artificial scenes like the Arcade Learning Environment. The second assumption of candidate model classes is a stronger assumption, but a large set of models could be defined directly given the object classes and only incur a cost quadratic in the set of features.

While encouraging, such results should be mostly viewed as a case study. We believe the key contributions of our paper are not this particular demonstration on *Pitfall!*, but rather the investigation of exploration approaches when planning can only be done approximately, and the benefit of selecting among model representations to support computationally tractable planning and fast learning. The second can be viewed as a bias/variance tradeoff and in future work we plan to consider how to identify and account for model biases that could limit asymptotic performance during the model-based planning procedure.

## 2 PRELIMINARIES

We consider a finite horizon Markov Decision Process (MDP) $\langle \mathcal{S}, \mathcal{A}, T, R, \gamma \rangle$, where $\mathcal{S}$ is the state space, $\mathcal{A}$ the action space, $T : \mathcal{S} \times \mathcal{A} \to \mathcal{S}$ the transition function, $R : \mathcal{S} \times \mathcal{A} \to \mathbb{R}$ the reward function, and $\gamma$ the discount factor. The goal of the RL agent is to maximize the expected discounted reward $\mathbb{E}_\pi[\sum_{t=0}^{T} \gamma^t R(s_t, a_t)]$ following a policy $\pi$.

## 3 APPROXIMATE MODEL-BASED PLANNING TO SUPPORT EXPLORATION

Computing an optimal policy for a Markov decision process in a large state space can be extremely computationally expensive, if not impossible. Therefore some form of approximate planning procedure is almost always needed for large scale domains. Especially in model-based RL algorithms when the transition and reward models are initially unknown and should be learned online , a computationally tractable planning algorithm is more critical. Since the model estimates are changing and one should plan more frequently with the new model estimates.

One popular and powerful approach for scaling up planning for large Markov decision processes is Monte Carlo Tree Search (Chaslot et al., 2008; Browne et al., 2012), which has been used to achieve better than world class performance in Go (Silver et al., 2016). When a perfect transition and reward model is known, MCTS is guaranteed to converge to the optimal value function in the limit of infinite

| Method | Optimism | Thompson Sampling | BAMCP |
|---|---|---|---|
| **Number of Episdoes** | $5.0 \pm 1.7$ | $6.2 \pm 1.95$ | $11.0 \pm 2.5$ |

Table 1: Comparison of different exploration methods in *mini-Pitfall!* to consistently achieve the reward at the right end of the first room.

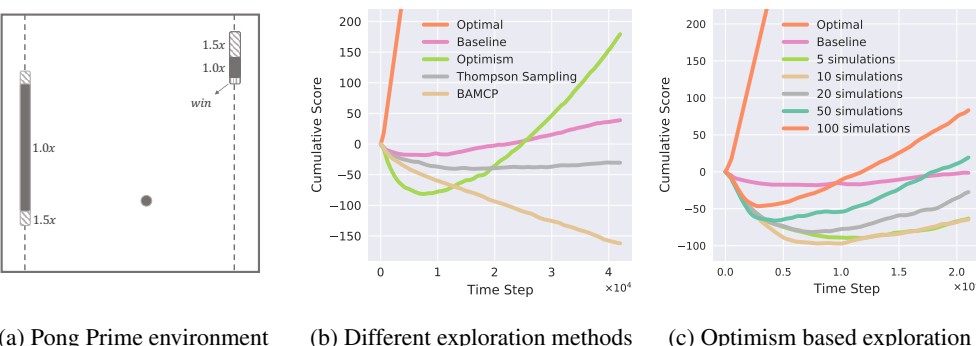

(a) Pong Prime environment  (b) Different exploration methods  (c) Optimism based exploration

Figure 1: Pong Prime

computation (i.e. infinite number of rollouts). However, when a model is estimated from the data and computational power is bounded, prior works has suggested how to adjust planning horizon to get the best computational performance (Jiang et al., 2015). As of our knowledge there is no work that additionally investigate the performance of different exploration methods with approximate planning.

We are interested in how to leverage models to support deep efficient exploration in large domains. Prior research has strong guarantees to find a near optimal policy with deep exploration when exact planning can be done, often in small state space (Brunskill et al., 2009; Brafman & Tennenholtz, 2002). In contrast, in this work, we investigate how to leverage models to support deep exploration when exact planning is not possible (large state space).

We propose a new optimistic MCTS to support deep exploration guided by models. Particularly, we suggest to perform a MCTS algorithm (e.g. UCT (Kocsis & Szepesvári, 2006)), and use the learned reward $R$ and transition model $T$ that is optimistic toward unseen or less frequently seen part of the state-action space (we further discuss how to learn simple models for planning in section 4). Concretely, while performing rollouts using learned models at each step an optimistic reward bonus is given. Optimistic reward bonus can be given by any optimism in the face of uncertainty (OFU) algorithm (e.g. MBIE-EB (Strehl & Littman, 2008) or Rmax (Brafman & Tennenholtz, 2002)).

Optimistic MCTS may have some advantages over alternate ways to achieve deep exploration in large state spaces given limited computation. In particular optimistic MCTS is leveraging model uncertainty to drive deep exploration, in contrast to policy search methods with simulated models (Sutton et al., 2000) that relies on the stochasticity of the policy for exploration and it is unclear how to leverage model uncertainty.

Additionally, MCTS with posterior sampling methods, like Thompson Sampling (Thompson, 1933) (by sampling a model from posterior distribution and performing rollouts) has strong guarantees when exact planning can be done, might not be optimistic enough for efficient exploration. With limited number of rollouts, agent might not observe the optimistic part of the model, in contrast to optimistic MCTS where optimism is built into every node of the tree.

To investigate the impact of approximate planning on deep exploration, we compare optimistic MCTS, with Thomson Sampling, and BAMCP (Guez et al., 2012) (a tractable sample-based method for approximate Bayes-optimal planning) in two toy domains, *Pong Prime* and *mini-Pitfall!*.

## 3.1 EXPERIMENTS

In order to test our hypotheses on the impact of imperfect model-based planning on deep exploration we introduced two toy environments, *PongPrime* and *mini-Pitfall!* that are similar to general Atari games but in smaller scale. *Pong Prime* environment is designed for a hard exploration task. Dynamics of this game is similar to Atari2600's Pong environment (Bellemare et al., 2013) with minor tweaks that make the game significantly harder. The enemy paddle is made 3 times larger than the player paddle. Additionally, the top and bottom 10% percent of the enemy paddle hit the ball back at 1.5 times the normal speed. Similarly, the player paddle also consists of 3 regions with distinct behavior. The *top region* of the paddle hits the ball back at 1.5 times speed. The *middle region* hits the ball back at normal speed. Finally, the *lower region* covers 5% of the paddle and instantly wins a point for the player. This configuration is set up so that it is difficult but not impossible for the player to score using the top region (scoring on average around 5% of the time the ball bounces off the top region). In this setting, the optimal policy is to always hit the ball with the lower region of the ball. The game is deterministic and model free methods with $\epsilon$-greedy exploration (e.g. DQN) consistently loses the game with lowest possible score across 5000 episodes. Figure1(a) shows this environment.

*mini-Pitfall!* is a small version of Atari2600 game *Pitfall!* which we use as a final test bed of our algorithm. In this version we limit the agent to two rooms of the game (the initial room, room 0, that agents starts the game in, and the room on the left side, room -1). There exist a dummy reward $R_{max}$ at the right end of the room 0, and the left end of room -1 is a terminal state (underground connection is also terminal state). Figure 2(a) shows this environment.

We provided the right model class for both experiments so we can separate the effect of exploration from model mismatch. Figure 1(b) compares the performance of different exploration strategies to the baseline (which uses a MLE model with UCT algorithm) combining with UCT algorithm in *Pong Prime* domain. We perform 500 total tree searches for all runs in Figure 1(b). Additionally, Table 1 shows the performance of different exploration methods in achieving the reward on the right side of the room 0 in *mini-Pitfall!* consistently.

## 3.2 DISCUSSION

Both BAMCP and TS perform worse than the MLE model. In the limit of infinite simulations, BAMCP is guaranteed to converge to the Bayes optimal solution (Guez et al., 2012). Similarly, with full horizon planning, we should be able to compute the exact value for the model sampled with TS, and there are strong guarantees that such a method will converge to the optimal policy (Osband & Van Roy, 2016). However, if it is infeasible to use a depth that mimics the game horizon, or perhaps even to reach a local reward, then TS may suffer. This is because TS samples a single model, which means that parts of the model may be overly optimistic, while other parts may be pessimistic. Hence, when performing a limited number of simulations using UCT, we may not go down branches of the tree that *observe* the optimistic parts of the sampled model. Therefore, the computed estimates of the Q value at the root node may not be optimistic, which is often a key part of proofs of the effectiveness of TS methods, and very helpful empirically.

Additionally, BAMCP suffers more in these environments that are deterministic. This means that for TS, optimism, and MLE approaches, the tree constructed will only have one child node (the deterministic next state) for any chosen action. In contrast, BAMCP samples a different deterministic model at each rollout, and for the same action node, those models may each deterministically predict different next states. Hence, BAMCP with $M$ sampled models and planning horizon $H$, potentially builds a tree of size $O((|A|M)^H)$, in contrast to the other methods that build a tree of at most size $O(|A|^H)$, where $|A|$ is the size of action space.

Optimism-based exploration significantly outperforms other approaches. Optimism is built into *every* node of the three that is allowing it to distinguish even locally between actions that may need exploration, in absence of observing long delayed reward. As we demonstrate in Figure 1(c) for the optimistic method, as planning power increases through more simulations (number of rollouts), the performance of optimism-based exploration also increases. With sufficient computational power, optimistic MCTS should learn the optimal policy for *Pong Prime* domain.

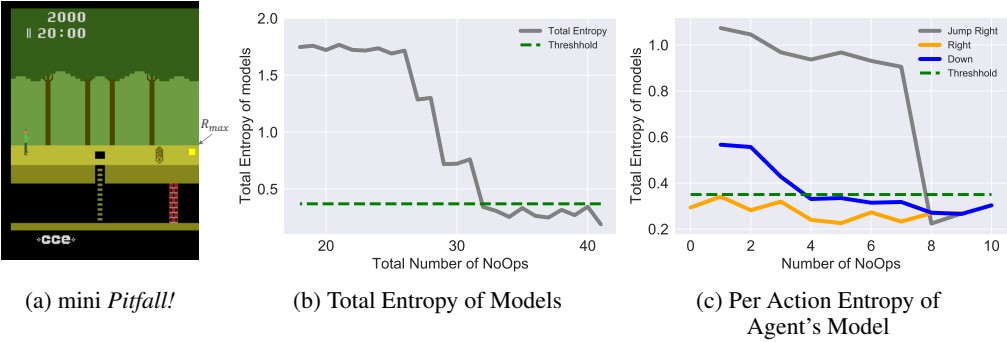

(a) mini *Pitfall!*  (b) Total Entropy of Models  (c) Per Action Entropy of Agent's Model

Figure 2: Macro actions

# 4 APPROXIMATELY DETERMINISTIC MODEL TO SUPPORT FAST LEARNING

In the previous section we considered how to plan and perform deep exploration given a transition and reward models. Now we ask a natural question, what types of model to learn?, and how these models will affect computational tractability of planning and sample efficient learning? In general simpler models are easier to learn and requires less data to train, in contrast to complex function approximation methods that often requires massive amount of data and suffer from compounding errors in lookahead planning (Roderick et al., 2017; Weber et al., 2017). Additionally, simple models are tractable to perform deep exploration with model uncertainty and reduce planning time.

There are reasonable evidence that humans learns a simple model, often inaccurate, to support planning and guide decision making (Tsividis et al., 2017; Dubey et al., 2018). In particular, people seem to benefit from higher level object representations that allows them to factor a high dimensional state space into simple low dimensional object states, this allows human to generalize from few examples, explore and plan efficiently. Additionally, as discussed in section 3.2, deterministic models can significantly help in planning and deep exploration, due to smaller branching factor of the tree (each state node will have only one child per action). Inspired by humans and prior works on object oriented MDP (Diuk et al., 2008) we hypothesize that object oriented models can help us learn a simple, easy and approximately deterministic models that are sufficient for planning .

Object detection has been long studied in computer vision, and state-of-the-art algorithms can detect objects with great accuracy in real world scenes (e.g. YoLo (Redmon & Farhadi, 2017), fast RCNN (Girshick, 2015) mask RCNN (He et al., 2017) and ...). We expect that these algorithms can simply detect objects and their bounding boxes, when they are trained on Atari2600. Thus, here we assume objects are given, in section 4.2 we discuss how to learn a simple model and further in section 4.3 we show how temporal abstraction can help learning an approximately deterministic mode.

## 4.1 REVIEW OF OBJECT ORIENTED MDP

We use a simpler version of OOMDP (Diuk et al., 2008). we define a set of object classes $\mathcal{C} = \{c_1, \ldots, c_n\}$ where each class has a set of attributes $\{c.a_1, \ldots, c.a_m\}$. Each state $s$ consists of objects $f(s) = \{o_1, \ldots, o_k\}$ where each object $o_i \in \mathcal{C}$. The state of an object is defined by the value assignment to its attributes. Finally, the state $s$ of the underlying MDP is the union of all object states $\cup_{i=1}^k o_i$.

We define the interaction function $I : \mathcal{O} \times \mathcal{O} \to \{0, 1\}$ to be an indicator that determines if two objects are interacting with each other. For simplicity, we make three assumptions: first, that this interaction function is known; second, objects from the same class share the same transition function; and third, each object's next state is dependent on at most pairwise object interactions and action. An object's successor state is determined by a standalone transition function $T_c(o, a)$ or a pairwise transition function $T_{c_i, c_j}(o_i, o_j, a)$ if $I(o_i, o_j) = 1$.

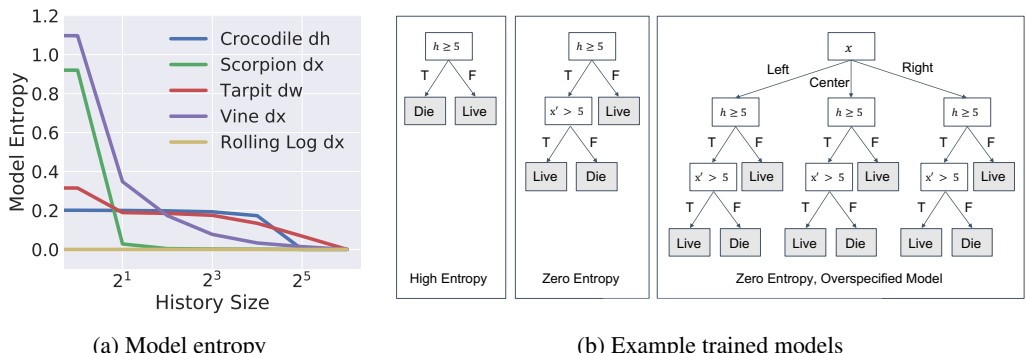

(a) Model entropy           (b) Example trained models

Figure 3: Model selection

## 4.2 MODEL LEARNING

Given that we are planning at an object level, we hypothesize that even simple models, such as linear and discrete count based models, give sufficient accuracy for planning, since often objects follow a very simple physical laws. More importantly, to ensure "sufficient accuracy in planning", we further require that these models predict transitions and rewards in a deterministic fashion.

To ensure deterministic transitions, we consider the class of functions $\mathcal{F}_t = \{f_t^1, \ldots, f_t^n\}$, where each $f_t^i$ is a count-based model of the dynamics for an object. Each function stores the count of every output based on a different set of input features with given history $t$. The simplest model in $\mathcal{F}_1$ is $f_1^1$, which uses one history with null input. For example, for a falling object with steady state velocity, such a model is sufficient as we can predict displacement $\delta x$ and $\delta y$ without any input. On the other hand, $f_1^n$, which uses one history and the most complex set of features, is the most complicated model in the class $\mathcal{F}_1$. In terms of objects, the most complex set of input features that we consider is the union of the object's state features, relative state features with respect to an interacting object, and action.

The goal then is to choose the simplest model that achieves deterministic transitions within $\mathcal{F}_t$. To do so, we compute the entropy of the observed data for each function $H(f_t^k) = -\sum_{x_i} p(x_i) \log(p(x_i|f_t^k))$ where the summation is over all the observed data. We choose the simplest model that has entropy less than a predefined threshold $\epsilon_{ent}$. If none of the models in $\mathcal{F}_t$ satisfy the entropy threshold, we increase $t$ through an exponential back off scheme. Concretely, we increase the history to the next exponent of 2. We use the same approach and same class of models for reward functions.

Figure 3(a) shows an example for the game *Pitfall!*, where we used a Cartesian product of object size $(w, h)$, object location $(x, y)$ and object intersection $(x', y')$ as features. Ignoring null input, this Cartesian product results in 7 different feature sets. With sufficient history, the entropy of all the models eventually drops to zero.

## 4.3 TEMPORAL ABSTRACTION

As we observed in our experiments, objects transitions, especially action-dependent transitions, can show a highly nonlinear behaviour and dependency to multiple time step histories. Inspired by human's *reaction time* and previous work (Diuk et al., 2008) we use the notion of macro action in the form of *"act and then wait"* in order to learn a simple approximately deterministic transition models.

Algorithm 1 shows the pseudo code to learn the macro actions from a predefined set of atomic actions (e.g. in the simplest form it can be the action space of the desired MDP; however, one can define them as any n-token combination of actions). In algorithm 1 we augment all atomic actions with $k$ number of *no-op* or wait, and then greedily decrease the number of no-ops followed by each action such that all model's can deterministically predict the next object state. The goal is to have a

model that achieves deterministic transition, to do so we measure the entropy of model's prediction with the observed data as in section 4.2.

---

**Algorithm 1: Macro Actions**

---

**Input:** maximum number of no-op $k$, set of atomic actions $\mathcal{A}$
macro actions $\leftarrow$ atomic action followed by maximum number of no-op;
mark all macro action as reducible;
**while** *there exist a reducible macro action* **do**
    $a \leftarrow$ select a reducible macro action and decrease number of no-op by one;
    $\tau \leftarrow$ compute entropy of model's prediction with new set of macro actions;
    **if** $\tau \geq thresh$ **then**
        mark $a$ as non reducible macro action and restore the number of no-op for $a$;
    **end**
**end**

---

Figure 2(b) shows the total entropy of model's versus total number of *no-op* in *Pitfall!* environment. As algorithm progress entropy increases to pass the threshold. Figure 2 (c) shows the entropy of agent's models while reducing number of *no-op* for one action and keeping others constant. As it shows Jump Right requires 8, Down requires 4 and Right requires none *no-op* afterward to achieves a deterministic transition for all actions. Running this algorithm in *Pong Prime* environment results in two *no-op* after each action, since the real dynamics uses 3 step history.

## 5 SOORL: STRATEGIC OBJECT ORIENTED REINFORCEMENT LEARNING

In this section we put together the insights from previous sections and propose a novel model-based object oriented RL algorithm, Strategic Object Oriented RL (**SOORL**). SOORL assumes access to an object detector, that returns a list of objects with their attributes (i.e. location and bounding box), an interaction function and macro actions (that can be learned with algorithm 1).

Algorithm 2 shows a pseudo code of SOORL. At each step, SOORL performs lookahead planning with UCT algorithm, learn and select appropriate transition and reward models for object representation and performs optimism based exploration.

---

**Algorithm 2: SOORL**

---

**Input:** object detector $f(s)$, lookahead planning depth $d$, number of rollouts $l$
initialize;
**for** *each episode e* **do**
    train value function V on replay buffer $\mathcal{D}$;
    **for** *each step i* **do**
        $o_i \leftarrow$ detect objects with object detector $f(s_i)$;
        $Q(s_i, a) \leftarrow$ perform lookahead planning with depth $d$ and $l$ rollouts;
        take action $a_i = argmaxQ(s,.)$ and update replay buffer $\mathcal{D} = \mathcal{D} \cup (o_i, a_i, o_{i+1}, r_i)$;
        update transition and reward models with $(o_i, a_i, o_{i+1}, r_i)$

---

**Model Learning:** SOORL uses the method described in section 4.2 with Cartesian product of object size $(w, h)$, object location $(x, y)$ and object intersection $(x', y')$ as features. Ignoring null input, this Cartesian product results in 7 different feature sets.

**Exploration:** Count based models allow SOORL to efficiently perform the knows what it knows (KWIK) (Li et al., 2008) scheme for exploration (optimistic MCTS). Concretely, if our algorithm queries the transition or reward model with a previously unseen input, we consider the resulting state as a state with $R_{max}$ reward. $R_{max}$ reward is also considered for any previously unseen object interactions. As we observe the reward for each interaction we update the reward model based on model-based interval estimation (Strehl & Littman, 2008).

**Planning:** At the beginning of each episode, a value function is trained based on previously seen transitions and rewards. Value function $V : \mathcal{O} \rightarrow \mathbb{R}$ is trained over object states and can generally be any function approximation methods.

Lookahead planning is performed by UCT (Kocsis & Szepesvári, 2006) algorithm with $l$ rollouts and depth $d$. Algorithm 3 shows pseudo code of planning, at each planning step, SOORL computes object interactions, selects appropriate models based on interactions and object states then predicts rewards and next object state. At depth $d$ of planning SOORL uses value of the object state $V(o_d)$ trained at the beginning of each episode.

---

**Algorithm 3: Lookahead Planning**

---

**Input:** objects $o$, depth $d$, rollouts $l$, value function $V$
**for** *number of rollouts $l$* **do**
 set current depth $i$ to zero;
 **while** $i \leq d$ **do**
  **if** $i == d$ **then**
   update $Q$ values with $V(o_i)$;
   break;
  detect interactions $I_i$ with objects $o_i$ and select models $T, R$;
  take action $a_i = argmaxQ(s_i, a) + c\sqrt{\frac{logN(s,a)}{N(s)}}$;
  obtain next object state $o_{i+1}$ and reward $r_i$ with $R(o_i, I_i), T(o_i, I_i)$ and update $Q$;

---

## 6   EMPIRICAL EVALUATION

In this section we will evaluate SOORL on an Atari game *Pitfall!*. SOORL assumes access to an object detector, a predefined set of function classes and macro action (that can be learned using algorithm described in section 4.3).

Labeled data for object's in Atari game in not available, and in our experiment we extracted the objects from Atari RAM and screen information. The need for an object detector makes engineering burden of SOORL prohibitive to test the algorithm on all Atari games. Thus we focused on one of the hardest game (sparse reward and hard exploration (Bellemare et al., 2016)), *Pitfall!*, where all the previous methods (without human demonstrations) failed to achieve any positive reward. We showed that object representation can be extremely helpful in this hard exploration game and SOORL can achieve a positive reward in *Pitfall!* without human demonstration.

*Pitfall!* is an Atari2600 environment where the goal is to have the agent traverse through multiple rooms (255 in total) while collecting rewards and avoiding obstacles. It is arguably one of the hardest Atari2600 game (Hester et al., 2017) due to its large map, sparse positive and dense negative rewards that necessitate deep exploration and long-horizon planning. The $\epsilon$-greedy exploration strategy completely fails in this environment, and more recent count-based exploration (Bellemare et al., 2016) does not show much performance boost due to the sparsity of positive reward. Pitfall is difficult even for human players without prior knowledge of the game – (Hester et al., 2017) reports that human performance varies from 3662 to 47821 points, whereas for other hard Atari games, this variation is much smaller (e.g. from 32300 to 34900 for Montezuma's revenge).

### 6.1   DETAILS

All objects information are extracted from RAM and screen information, and each object's attribute is location $(x, y)$ and bounding box size $(w, h)$. Objects are considered interacting with each other if bounding boxes collide. Transition and reward models for each object are based on the method described in Section 4.2. The features used are a Cartesian product of object size $(w, h)$, object location $(x, y)$ and object intersection $(x', y')$. Ignoring null input, this Cartesian product results in 7 different feature sets.

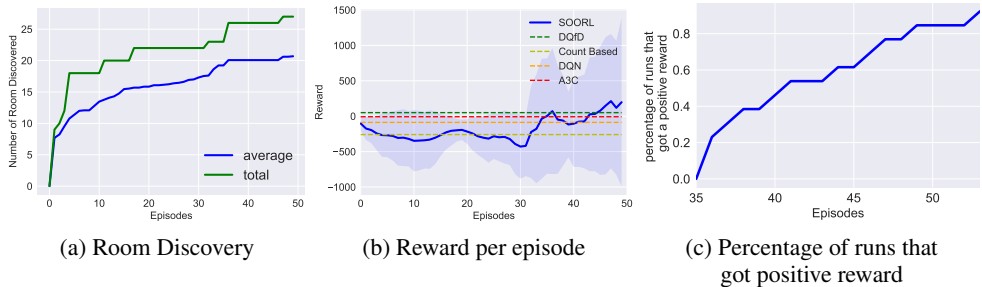

(a) Room Discovery      (b) Reward per episode      (c) Percentage of runs that got positive reward

Figure 4: performance of SOORL on Pitfall!

As described in section 3 we used optimism based exploration method, by assigning reward $R_{max}$ to all unseen interactions and transitions. As we observe reward for each interaction we update the model, based on model based interval estimation (Strehl & Littman, 2008). Additionally in order to further incentivize exploration we split the screen into $N \times M$ grids and keeping a count of the number of times agent visits each grid. The agent is given a reward bonus $\beta n(s)^{-1/2}$ based on visit count $n(s)$.

For value function, we used discrete object's location with the same split used for optimism bonus, and computed the empirical transition $\hat{T}(o'|o, a)$ and empirical reward with optimism bonus $\hat{r}(o, a) + \beta n(s)^{-1/2}$, and at the beginning of each episode, we performed value iteration.

## 6.2 PERFORMANCE AND DISCUSSION

Figure 4(a) shows an increasing number of rooms being discovered across episodes. On average, the agent discovers 21 rooms within 50 episodes, and in total across 20 runs the agent discovers 27 different rooms. This validates our hypothesis that optimistic MCTS drives deep exploration, that we also showed in a smaller domains in section 3.

Figure 4(b) shows accumulative reward for each episode of SOORL and compares them to the other state of the art algorithm. Results of count-based (Bellemare et al., 2016), DQfD (Hester et al., 2017), A3C (Mnih et al., 2016) and DQN (Mnih et al., 2015) are reported at the time of evaluation. Our average score across all episodes and all runs is $-193.5 \pm 595.8$ and SOORL score for the best episode across all runs in $606.6 \pm 1254.5$, which is higher than all scores that were reported at the time of evaluation. To the best of our knowledge, this is the first approach which manages to get positive rewards on *Pitfall!* without human demonstrations. Sample videos of the agent reaching the two closest positive rewards can be found here: `https://youtu.be/GvenPZMJiTg` (4000 reward) `https://youtu.be/74F-ta5LyuA` (2000 reward)

Figure 4 (c) shows the percentage of runs that got a positive reward. More than $80\%$ of runs got a positive reward in 50 episodes that shows consistency of our approach across multiple runs. Due to simple dynamics model that can be learned fast SOORL is extremely sample efficient in comparison to other deep RL method that often takes millions of frame to find a good policy. However, end-to-end deep RL methods use significantly less prior knowledge and using raw pixels as input, thus a direct comparison with them is unfair. We have also provided macro actions and object information for DQN and DDQN but those methods are not designed to take advantage of object representation and did not show a boost in performance. On the other hand, SOORL uses much less prior knowledge than methods with human demonstration (Aytar et al., 2018; Hester et al., 2017), where human guidance enormously reduce the challenge of exploration.

Additionally, macro actions are provided for SOORL, but as shown in section 4.3 these can be learned online. Integrating this temporal abstraction with SOORL can increase the sample complexity by $\mathcal{O}(|A|K)$, where $K$ is the maximum number of no-ops.

## 7 RELATED WORK

Model based RL has accomplished great success in tasks in which a perfect simulator is known (Silver et al., 2016; 2017), mostly using Monte Carlo Tree Search algorithms like UCT (Kocsis & Szepesvári, 2006; Chaslot, 2010). Recent work has focused on applying deep learning to model-based RL by learning the model online (Rosin, 2011; Finn & Levine, 2017; Lenz et al., 2015; Weber et al., 2017). In contrast to these methods, we seek to learn a simple model based on object representation.

Exploration has been extensively studied in the tabular setting (Brafman & Tennenholtz, 2002; Strehl & Littman, 2008; Jaksch et al., 2010; Osband & Van Roy, 2016). However, these methods do not scale well to large MDPs and often result in poor sample complexity. Additionally, these methods assume exact planning that might not be feasible in large state space. Recent approaches (Bellemare et al., 2016; Ostrovski et al., 2017; Tang et al., 2017) proposed an extension of count based exploration to large MDPs. Despite good asymptotic performance, when compared with humans (Tsividis et al., 2017; Lake et al., 2017), these methods require orders of magnitude more samples.

Bayesian RL methods also provide an effective balance between exploration and exploitation (Ghavamzadeh et al., 2015). When exact planning is possible, they provide strong guarantees. However, these methods remain computationally intractable in large state spaces. Recent works propose an extension of these methods to large MDPs (Osband et al., 2016; Azizzadenesheli et al., 2018; Fortunato et al., 2017). However, these methods are unable to show substantial improvement in hard exploration, sparse reward environments.

A closely related line of research, finite horizon planning (Kearns et al., 2002; Mannor et al., 2007; Kearns et al., 1994), has noted how planning horizon can affect planning loss (Kearns & Singh, 2002; Strehl et al., 2009). Recent study has shown that shorter planning horizons might be better when there is model inaccuracy (Jiang et al., 2015). In contrast, in this work, we studied how imperfect planning can affect exploration.

OOMDP (Diuk et al., 2008) defines a notion borrowed from relational MDPs (Guestrin et al., 2003), and uses objects to learn models and perform model based planning. Model free methods that use object representation (Garnelo et al., 2016; Roderick et al., 2017; Cobo et al., 2013) fail to scale to large MDPs and do not leverage object representation for deep exploration. The main difference between our approach and other object oriented approaches is that we perform scalable planning with strategic exploration by leveraging objects to learn simple dynamics models.

## 8 CONCLUSION AND FUTURE WORK

To conclude, we showed how we can achieve a sample efficient RL algorithm with object priors. We proposed optimistic MCTS as a way to drive deep exploration when exact planning is impossible, and showed this to be more effective than posterior sampling methods. Additionally, we investigate how approximately deterministic simple models can be learned with object representation to support fast learning and planning.

We introduced Strategic Object Oriented RL (SOORL) that uses object representation and optimistic MCTS with automatic model selection that biases towards simple deterministic models. SOORL achieves state of the art results in the game of *Pitfall!*. While there remains works to be done to reduce the engineering burden of SOORL, lookahead planning with object representation is a very promising path towards more sample efficient RL algorithms.

A very important line of future research is **robust planning**. One important challenge in model-based RL is making planning robust to model inaccuracy. Identifying the right model class is a nontrivial task, and a wrong model class can easily introduce a catastrophic error in long-horizon prediction that prohibits the use of tree search algorithms like UCT.

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
