# OpenReview forum: "Fast Exploration with Simplified Models and Approximately Optimistic Planning in Model Based Reinforcement Learning"
_ICLR.cc/2019/Conference_

### Official Review · AnonReviewer3 · 2018-11-04
**A heavily engineered approach which achieves good performance in limited settings**

**Rating:** 4
**Confidence:** 4

**Review:**

This paper proposes a model-based object-oriented algorithm, SOORL.
It assumes access to an object detector which returns a list of objects with their attributes, an interaction function which detects interactions between objects, and a set of high-level macro actions. Using a simplified state representation obtained through the object detector, it performs optimistic MCTS while simultaneously learning transition and reward models. The method is evaluated on two toy domains, PongPrime and miniPitfall, as well as the Atari game Pitfall. It achieves positive rewards on Pitfall, which previous methods have not been able to do.

Despite good experimental results on a notoriously hard Atari game, I believe this work has limited significance due to the high amount of prior knowledge/engineering it requires (the authors note that this is why they only evaluate on one Atari game). I think this would make a good workshop paper, but it's not clear that the contributions are fundamental or generally applicable to other domains. Also, the paper is difficult to follow (see below).

Pros:
- good performance on a difficult Atari game requiring exploration
- sample efficient method

Cons:
- paper is hard to follow
- approach is evaluated on few environments
- heavily engineered approach
- unclear whether gains are due to algorithm or prior knowledge


Specific Comments:

- Section 3 is hard to follow. The authors say that they are proposing a new optimistic MCTS algorithm to support deep exploration guided by models, but this algorithm is not described or written down explicitly anywhere. Is this the same as Algorithm 3 from Section 5? They say that at each step and optimistic reward bonus is given, but it's unclear which bonus this is (they mention several possibilities) or how it relates to standard MCTS.
In Section 3.1, it is unclear what the representation of the environment is. I'm guessing it is not pixels, but it is discrete states? A set of features?
The authors say "we provided the right model class for both experiments" - what is this model class?

- Concerning the general organization of the paper, it would be clearer to first present the algorithm (i.e. Section 5), go over the different components (model learning, learning macro actions, and planning), and then group all the experiments together in the same section.
The first set of experiments in Sections 3.1 and 3.2 can be presented within the experiments section as ablations.

- Although the performance on Pitfall is good, it's unclear how much gains are due to the algorithm and how much are due to the extra prior knowledge. It would be helpful to include comparisons with other methods which have access to the same prior knowledge, for example with DQN/A3C and  pseudo-count exploration bonuses using the same feature set and macro actions as SOORL uses.


Minor:
- Page 2: "Since the model...the new model estimates": should this be part of the previous sentence?
- Page 5: "There are reasonable evidence" -> "There is reasonable evidence"
- Page 5: ". we define a set of..." -> ". We define a set of..."
- Page 8: "any function approximation methods" -> "method"

---

### Official Review · AnonReviewer4 · 2018-11-09
**Good sample-efficient performance on Pitfall using planning and imperfect models, but with limited impact due to simplifications that are hard to remove/circumvent.**

**Rating:** 5
**Confidence:** 4

**Review:**

-- Summary --

The paper proposes to learn (transition) models (for MDPs) in terms of objects and their interactions. These models are effectively deterministic and are compatible with algorithms for planning with count-based exploration. The paper demonstrates the performance of one such planning method in toy tasks and in Pitfall, as well as a comparison with other planning methods in the toy tasks. The proposed model-based method, called SOORL, yields agents that perform better on Pitfall with a small amount of data.

-- Assessment --

As a positive, the results of the paper are favorable compared to previous work, with good sample efficiency, and they demonstrate the viability of the proposed approach. The most negative point is that SOORL relies on limiting domain-specific biases that are hard to remove or circumvent.

-- Clarity --

The paper is somewhat clear. There are many typos and mistakes in writing, and at parts (for example, the second paragraph of Section 4.2) the explanations are not clear.

-- Originality --

I believe the work is original. The paper explores a natural idea and the claims/results are not surprising, but as far as I am aware it has not been tried before.

-- Support --

The paper provides support for some of the claims made. The comparison to related work contains unsupported claims ("we studied how imperfect planning can affect exploration") and could be more upfront about the weaknesses of the proposed method. The claims in the introduction are sufficiently supported.

-- Significance --

It would be hard to scale SOORL to other tasks, so it is unlikely to be adopted where end-to-end learning is wanted. Therefore I believe the impact of the paper to be limited.

There is also the question of whether the paper will attract interest and people will work on addressing the limitations of SOORL. I would like to hear more from the authors on this point.

-- For the rebuttal --

My greatest doubt is whether the paper will attract enough interest if published, and it would be helpful to hear from the authors on why they think future work will build on the paper. Why is the proposed approach a step in the right direction?

-- Comments --

Sample efficiency: The paper should be more clear about this point. It seems that 50 episodes were used for getting the positive reward in Pitfall, which is great.

Object detection: I am happy with the motivation about how we can remove the hand-made object detection. It is important the other strong assumptions (object interaction matrix, for example) can be removed as well. My opinion on simplifications is this: They are ok if they are being used to make experiments viable and they can be removed when scaling up; but they are not ok if there is no clear way to remove them.

Known interaction matrix: It may be possible to remove this requirement using the tools in [1]

Deterministic model: The use of no-ops to make the model deterministic seems right if the ultimate goal is to make the model deterministic, but it seems unsuited if the model is to be used for control. Maybe the model needs to be temporally extended as I thought the paper was proposing in Section 4.2 but Section 4.3 suggests that this temporal extension was not a good idea. Is my understanding correct?

Exploration: I was a bit confused about how the text discusses exploration. UCT uses OFU, but the text suggests that it does not. What are the components for exploration? Both a bonus on unseen transitions and the confidence interval bonus? Also, the paper would have to provide support for the claim that "with limited number of rollouts, the agent might not observe the optimistic part of the model, in contrast to optimistic MCTS where optimism is build into every node of the tree". However, it is fair to say that in the to domains MCTS seemed has performed better, and for that reason it has been chosen instead of Thompson Sampling for the later experiments.

Writing: The paper has a number of typos and mistakes that need to be fixed. To point out a few:
* I would suggest more careful use of "much" and "very"
* For citations, "Diuk et al. (2008) also proposed..." and "(UCT, Kocsis & Szepesvari, 2006)"

Claims: I think the claims made in the introduction could be stated more clearly in the conclusion. (Intro) "We show how to do approximate planning" --> (Conclusion) "Our model learning produces effectively deterministic models that can then be used by usual planning algorithms".

-- References --

[1] Santoro et al., 2017. "A simple neural network module for relational reasoning"

---

### Meta-Review · Area_Chair1 · 2018-12-14

**Confidence:** 5
**Recommendation:** Reject

**Metareview:**

Pros:
- rather novel approach to using optimistic MCTS for exploration with deterministic models
- positive rewards on Pitfall

Cons:
- lost of domain-specific knowledge
- deteministic models
- lacking clarity
- lacking ablations
- no rebuttal

I agree with both reviewers that the paper is not good enough to be accepted.